# Impact Performance and Bending Behavior of Carbon-Fiber Foam-Core Sandwich Composite Structures in Cold Arctic Temperature

**M.H. Khan [1], Bing Li [2] and K.T. Tan [1],\***

[1]  Department of Mechanical Engineering, The University of Akron, Akron, OH 44325-3903, USA; mhk11@zips.uakron.edu

[2]  School of Aeronautics, Northwestern Polytechnical University, Xi'an 710072, China; bingli@nwpu.edu.cn

\*  Correspondence: ktan@uakron.edu; Tel.: +1-330-972-7184; Fax: +1-330-972-6027

**Abstract:** This study investigates the impact performance, post-impact bending behavior and damage mechanisms of Divinycell H-100 foam core with woven carbon fiber reinforced polymer (CFRP) face sheets sandwich panel in cold temperature Arctic conditions. Low-velocity impact tests were performed at 23, −30 and −70 °C. Results indicate that exposure to low temperature reduces impact damage tolerance significantly. X-ray microcomputed tomography is utilized to reveal damage modes such as matrix cracking, delamination and fiber breakage on the CFRP face sheet, as well as core crushing, core shearing and debonding in the Polyvinyl Chloride (PVC) foam core. Post-impact bending tests reveal that residual flexural properties are more sensitive to the in-plane compressive property of the CFRP face sheet than the tensile property. Specifically, the degradation of flexural strength strongly depends on pre-existing impact damage and temperature conditions. Statistical analyses based on this study are employed to show that flexural performance is dominantly governed by face sheet thickness and pre-bending impact energy.

**Keywords:** composite sandwich structure; impact performance; bending behavior; arctic temperature

---

## 1. Introduction

Reduction in Arctic sea ice in the region over the last three decades has opened more efficient sailing routes [1]. New seaways through the Northern route have resulted in the increased deployment of marine and naval vessels in extreme low-temperature Arctic conditions. This requires advanced materials to combat the fundamental challenges associated with operating in such a cold and harsh environment. Therefore, a better understanding of how materials and structures behave and perform at extremely low temperatures is of utmost importance.

Advanced composite materials are widely used in automotive, aerospace, wind energy and marine industries. Sandwich composites are commonly employed in aircraft structures, ship hulls, wind turbine blades and bridge decks due to their enhanced bending stiffness, low weight, excellent thermal insulation, and acoustic damping capabilities. Sandwich structures typically consist of two thin and stiff skins (face sheets), which are separated by a thick, light, and shear-resistant core. Foam core sandwich structures are preferred to honeycomb structure for marine applications due to their low water absorption properties. However, one of the major concerns in the use of sandwich composites is their susceptibility to impact damage, which may occur during service and maintenance conditions. Impact loadings like bird strike, tool drop, hailstones, and debris impact by hurricane or tsunami can significantly reduce residual strength of the sandwich composites [2]. Low-velocity impact (less than 10 m/s) is commonly used to analyze the impact-induced damage phenomenon, because damage

incurred is barely visible, yet extremely detrimental to the post-impact health of the composite structure [3]. Raju et al. [4] and Xue et al. [5] studied how the thickness of a honeycomb core affected the impact tolerance of sandwich panels. Atas and Potoglu [6] and James et al. [7] examined how the carbon fiber reinforced polymer (CFRP) thickness improved low-velocity impact resistance composite structures, and confirmed that improvement to impact damage can be made by using a thicker core and face sheet. Zhang and Tan [8], Huo et al. [9], Xin et al. [10] investigated how the shape of the impactor head affected the impact performance of sandwich composites. Papa et al. [11] studied the impact resistance and flexural behavior dependency on the stacking sequence of fiber hybrid composites. A constitutive model was proposed to understand the failure features and strain rate dependency of composite structures by Long et al. [12].

Jia et al. [13] found out that CFRP composites have enhanced flexural strength, maximum deflection, and energy absorption at a lower temperature. However, this study was limited only to composite laminates. Khan et al. [14] investigated impact performance and damage modes of Polyvinyl Chloride (PVC) foam core sandwich structures subjected to low-velocity impact damage. However, the study of post-impact structural integrity and damage mechanisms associated with after-impact loading was lacking. Schubel et al. [15] studied low-velocity impact and the post-impact compressive strength of composite sandwich panels at room temperature. However, the work lacked any study of how post-impact loading, such as bending, influenced the damage tolerance of sandwich structures in cold temperature Arctic conditions.

Yang et al. [16] explored how temperature influences impact behavior in foam-core sandwich composites under low-velocity range and concluded that low temperature results in reduced damage area and indentation depth. However, the understanding of important features with respect to impact resistance such as peak force and energy absorption was not analyzed. Erickson et al. [17] studied the effect of temperature on low-velocity impact and the post-impact bending behavior of composite sandwich panels. Although the low-velocity impact tests were executed at different temperatures, the post-impact bending tests were performed constantly at room temperature. This approach has failed to provide a good understanding of the relationship between temperature and post-impact bending performance.

In this study, low-velocity impact response, impact induced damage mechanisms and the post-impact flexural behavior of carbon-fiber foam-core sandwich composites are investigated at room and low temperature conditions. Section 1 provides an introduction, covering related published papers and their research gaps that have motivated the current work. Section 2 describes the materials and methods used in the current investigation, including the impact test setup, X-ray tomography technique, three-point bending test setup, and the statistical design of experiment approach. Section 3 presents the results and discussion, in terms of impact performance, impact damage mechanisms, residual flexural strength after impact, post-bending micro-computed tomography, and statistical analysis for understanding the influence of the factors. Section 4 ends with a conclusion for the current work. This work provides a better understanding of the impact dynamic response of composite sandwich structures at extremely low Arctic temperatures.

## 2. Materials and Methods

### 2.1. Materials

The sandwich composite is made of Divinycell Polyvinyl Chloride (PVC) H-100 foam core (6.35 mm thick) between 0/90° woven carbon fiber reinforced polymer (CFRP) face sheets. The lay-up schedule of the sandwich composite is 3 layers of plain weave carbon fiber, Divinycell H-100 Foam and another 3 layers of plain weave carbon fiber. A wet layup process was used and co-cured all together, peel ply was put down on tool, each fabric ply is laid down with resin applied, then core, then top face sheet plies with resin. Breather ply, caul plate and part was vacuum bagged. After vacuum was pulled, the specimen was cured in the oven, debagged and trimmed to final size. Epoxy resin accounted for

approximately 50% of the composition. As the specimens are made by DragonPlate company, the exact epoxy resin and fiber types remain proprietary.

Two types of specimens were tested: thinner face sheet (only one woven CFRP lamina) and thicker face sheet (three woven laminae) on each skin. Each lamina was 0.25 mm in thickness. The sandwich panel was then cut into 150 mm × 100 mm sample size. For low-temperature testing, specimens were first cooled down to −23 °C in a freezer over a day period. −23 °C was the lowest temperature that the freezer could achieve. Subsequently, before impact testing, they were further cooled to −30 and −70 °C in the environmental chamber of an Instron CEAST 9350 impact machine using liquid nitrogen gas. The preconditioning in the freezer allowed the specimens to cool gradually from room temperature to low temperature, so that the specimens did not experience rapid cooling that might have caused damage to the specimens due to sudden thermal shrinkage.

### 2.2. Impact Test

The Instron CEAST 9350 drop tower, having a 16 mm diameter hemispherical impactor as shown in Figure 1, was used for the impact. The specimen was placed on a cylindrical support frame with a window of 76 mm diameter. Specimen was clamped with 100 N clamping force to avoid any tilt or distortion during impact. A 3.482 kg striker was raised to a height of 0.234 m and 0.117 m and subsequently dropped with impact velocities of 2.14 m/s and 1.52 m/s, respectively, to generate 8 J and 4 J impact energy. These energy levels were chosen such that the lower energy of 4 J damaged only the front face sheet, and 8 J impact penetrated the front face sheet and incurred damage to the core. Strain gauges were present inside the tup that was connected to the steel impactor. The strain measurements over the duration of the impact event provided the displacement of the impactor (deflection of the specimen), which were subsequently integrated to calculate the velocity and acceleration profiles, and thereby the force measurements. All these data were captured and measured by the DAS64K data acquisition system of Instron CEAST 9350 impact machine. Specimens were tested at 23, −30 and −70 °C. The temperature of 23 °C was selected as the benchmark case for room temperature, while −30 and −70 °C are the average and lowest temperature in the Arctic region, respectively [18]. A thermostatic chamber cooled by liquid nitrogen was used for low-temperature testing. At least four specimens were tested for each case.

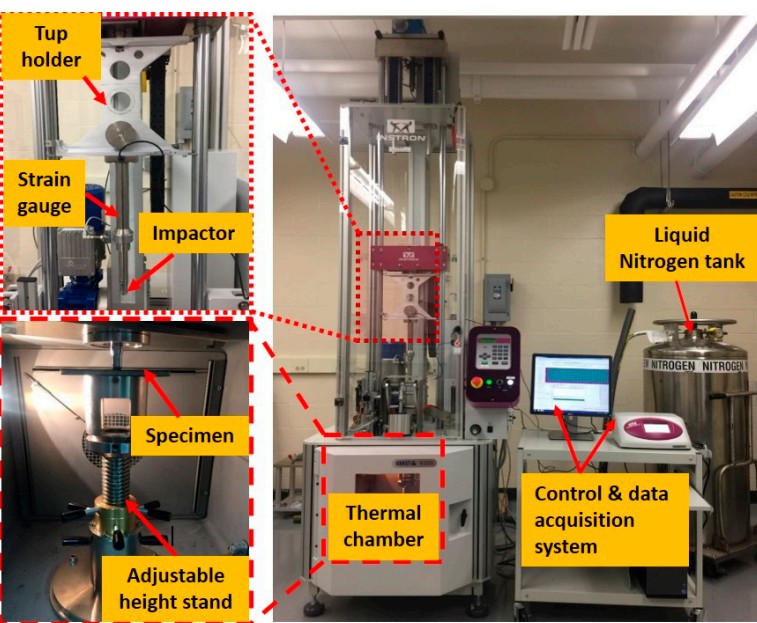

**Figure 1.** Experimental setup for impact test with Instron CEAST 9350 drop tower that is equipped with a thermal chamber for low-temperature testing and a control and data acquisition system to capture force, time, displacement and energy impact data.

### 2.3. X-ray Tomography for Damage Inspection

In this study, 3D renderings and cross-sectional images of the samples were acquired for both post-impact, and post-bending test specimens using a Nikon Metrology XTH 320 LC X-ray micro-computed tomography system. Numerous projection images were captured, as the sample rotated a complete 360° revolution. A 3D reconstruction was done using software CT Pro provided by Nikon Metrology. A 225 kV microfocus X-ray source penetrated the specimen for tomography with X-ray emission scanning at 90 keV and 50 µA. Each rotational image was averaged twice, acquiring 1800 scans. The collected data was 41.7 µm in voxel size. VG Studio Max software was then used for 3D image reconstruction.

### 2.4. Post-Impact Three-Point Bending Test

Post-impact bending tests were conducted to investigate the residual flexural strength of sandwich structures as shown in Figure 2, using the Instron 5582 machine following ASTM C 393 standard with a crosshead speed of 0.5 mm/min. Samples were tested at a room temperature of 23 °C and cold temperatures of −30 and −70 °C using an environmental chamber cooled by liquid nitrogen. The specimen was tested such that the impacted face sheet experienced either compression (inward) or tension (outward). These orientations were chosen to resemble actual service conditions, whereby depending on the supports, the composite structure could either bend concave or convex after being subjected to impact. In all cases, at least two specimens were tested under each condition.

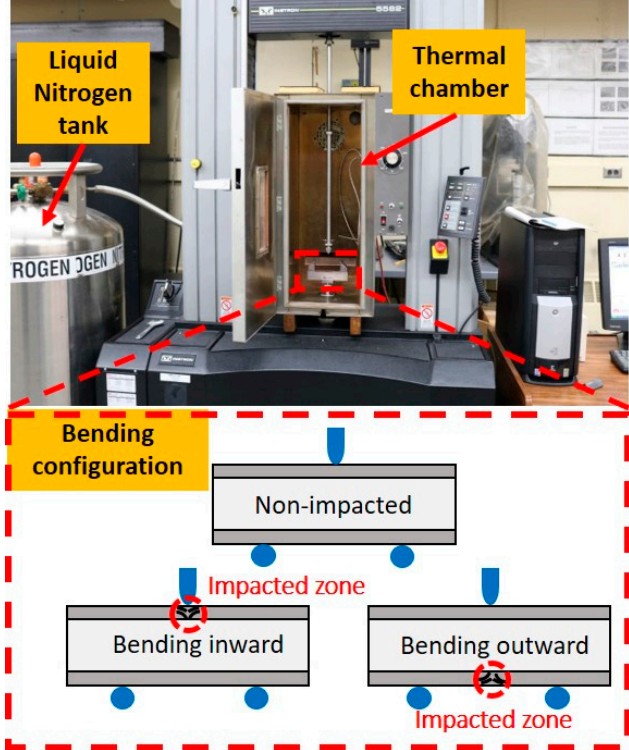

**Figure 2.** Experimental setup for the bending test with an Instron 5582 universal testing machine that is equipped with thermal chamber for low-temperature static testing. Schematic illustrates three bending configurations during bending test: nonimpacted case; bending inward case where impact damage is at top face sheet; bending outward case where impact damage is at bottom face sheet.

### 2.5. Statistical Design of Experiment

Design of Experiments (DoE) is a statistical tool that defines how input factors influence an outcome [19–21]. Standard factorial design dictates all variables having the same number of levels. In this study, two face sheet thickness (0.25 mm and 0.75 mm); three different temperatures (23, −30 and

−70 °C); two impact energies (0 J and 8 J); two bending configurations (bending inward and outward), were considered as the main factors. Table 1 presents the two-level DoE used for the four factors. The aim is to understand how these factors affect after-impact flexural performances. The 0 J impact energy was considered as the nonimpacted bending configuration and 4 J and 8 J were considered the impacted bending configurations. For each configuration of nonimpacted, inward and outward cases, testing was repeated at least twice and the peak force values were tabulated.

**Table 1.** Four factorial two levels Design of Experiment (DoE).

| Factors | Factor Levels | |
| --- | --- | --- |
| | Low Level (−1) | High Level (+1) |
| Face sheet thickness | 0.25 mm | 0.75 mm |
| Temperature | −70 °C | 23 °C |
| Impact energy | 0 J | 8 J |
| Bending configuration | Inward | Outward |

## 3. Results and Discussion

### 3.1. Impact Performance

Representative force-displacement plots for both thick and thin specimens impacted with 4 J and 8 J at different temperatures 23, −30, and −70 °C are presented in Figure 3. The sampling rate used during the impact test was 100 kHz for data acquisition, which gave sufficient precision and accuracy in the measurement of low-velocity impact tests. The markers in Figure 3 are intentionally spaced out at fixed intervals to make the curves distinguishable for different temperatures. The force-displacement curves showed linear behavior up to a certain point (defined as critical force for front face sheet damage) whereby sudden load drop indicated the failure of the front carbon fiber/epoxy composite face sheet (Figure 3a–c), except in a thick specimen impacted with low and insufficient impact energy of 4 J (Figure 3d), in which significant recovery happened along the displacement axis due to rebound of the impactor. However, for thin specimens, 4 J energy was enough to fail the front face sheet (Figure 3b). The critical forces for front face sheet penetration are shown in Figure 4. As the temperature decreased the force required for front face sheet penetration decreased for both thick and thin specimens due to the increased brittleness of the carbon fiber reinforced polymer (CFRP) face sheet. Low temperature severely degraded the composite face sheet's impact tolerance, the epoxy matrix of face sheet became exceptionally brittle, thus requiring less force for matrix cracks and brittle fracture [14,15]. A thicker face sheet offered higher resistance than a thin face sheet [5], thus thin specimen front sheets were penetrated at lower force than thick specimens. After the sudden load drop, as seen in Figure 3a–c, impact force continued to rise over time due to foam densification and finally declined until the end of the impact event with significant permanent deformation (Figure 3a–c) and rebound (3d). Moreover, with a decrease in temperature, the permanent displacement induced by impact for thin (4 J and 8 J) and 8 J for thick specimens of force-displacement plots generally increased, indicating more damage at lower cold temperatures. The 4 J thick specimen did not show this trend because the impact did not induce sufficient damage to the front face sheet.

The area under the curve in Figure 3 represents the energy absorbed by the damaged specimen and is shown in Figure 5. With the increase of impact energy, Figure 5a, and a decrease in temperature, Figure 5b, the percentage of energy absorbed increased, thereby indicating greater damage induced by impact at low temperature. The absorbed energy was also more sensitive to temperature change at low impact energy level (4 J) compared to higher impact energy (8 J) (Figure 5b) where absorbed energy had reached a high consistent percentage of more than 90%.

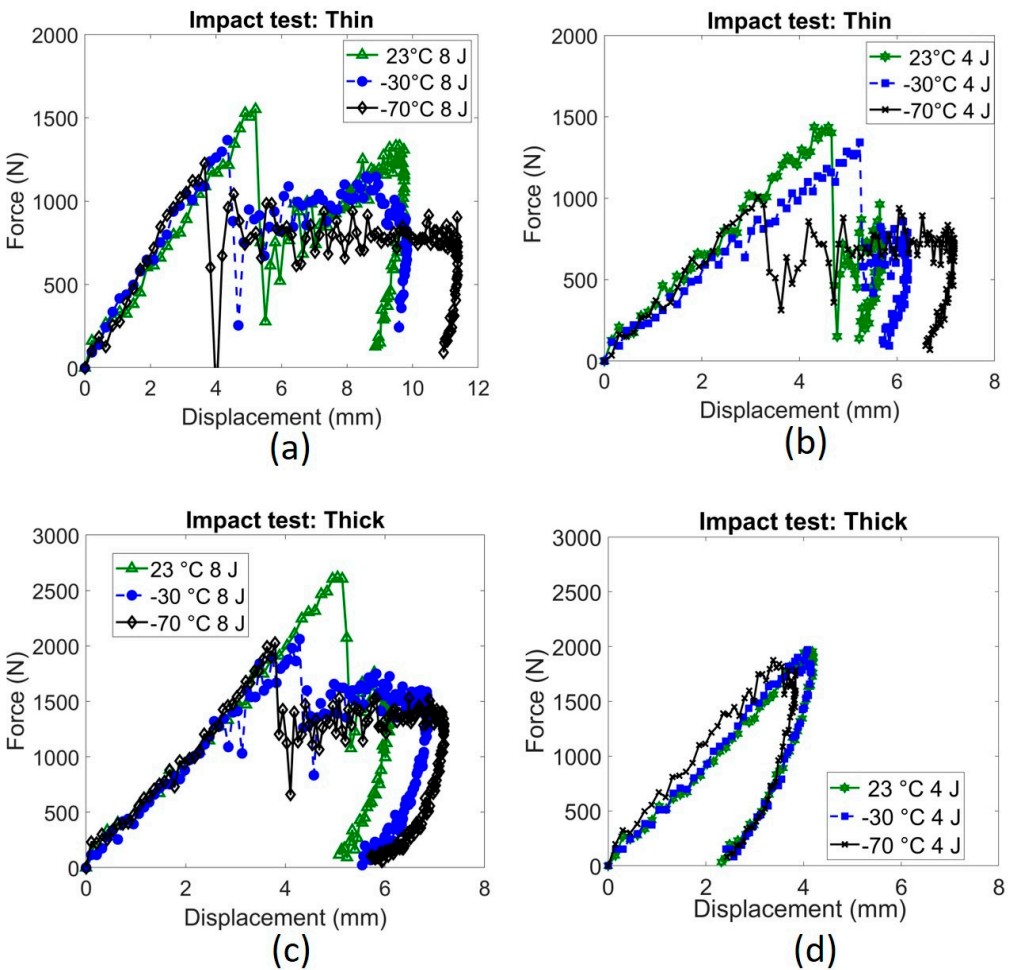

**Figure 3.** Impact force-displacement plots for (**a**) thin 8 J; (**b**) thin 4 J; (**c**) thick 8 J; (**d**) thick 4 J.

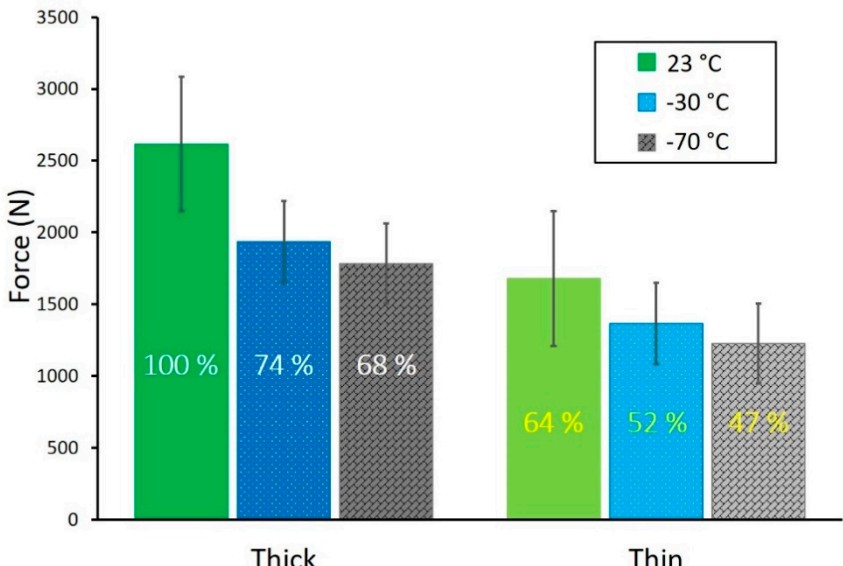

**Figure 4.** Critical front face sheet damage force for 8 J impact energy.

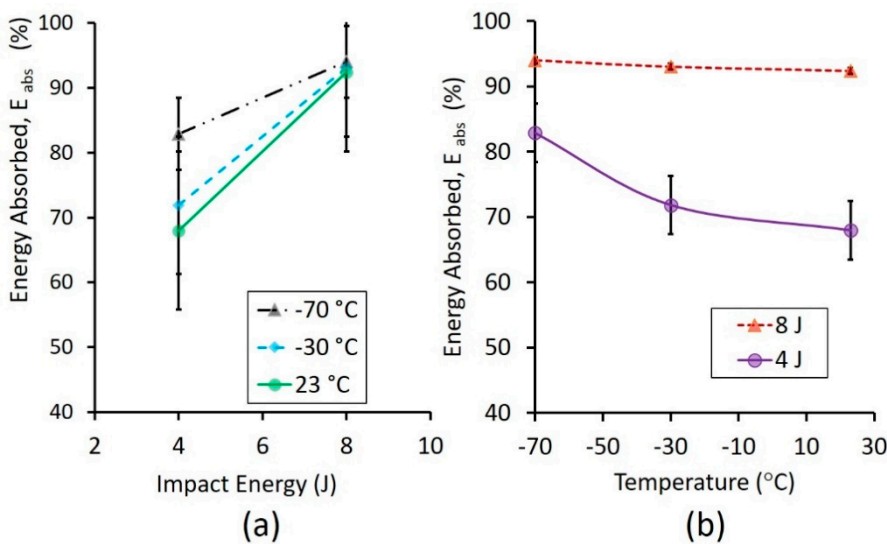

**Figure 5.** Thick specimens impacted with specified impact energy: (**a**) energy absorbed against impact energy; (**b**) energy absorbed against test temperature.

### 3.2. Impact Damage Mechanisms

Visible impact damage for thin specimens impacted with 8 J impact energy are shown in Figure 6. At 23 °C (Figure 6a), the damage was limited to localized matrix cracking only; while at −30 °C, matrix cracks were propagated deeper and longer both vertically and horizontally along the 0/90° woven fiber directions of the face sheet (Figure 6b). At the most extreme condition of −70 °C (Figure 6c), severe damage of matrix cracks and fiber breakage occurred, indicating face sheet penetration. X-ray μCT images are exhibited in Figure 7. At 23 °C (Figure 7a) the front face sheet suffered delamination and matrix failure, though partial penetration was observable. This ultimately led to fiber breakage at the front face sheet. Beyond front face sheet penetration, foam core damage was characterized by two mechanisms: core densification and core shearing. Core densification occurs due to high compressive normal stress loading under the impact region. Core densification is induced either by front face sheet compression to the core or by impactor tip compressing the core.

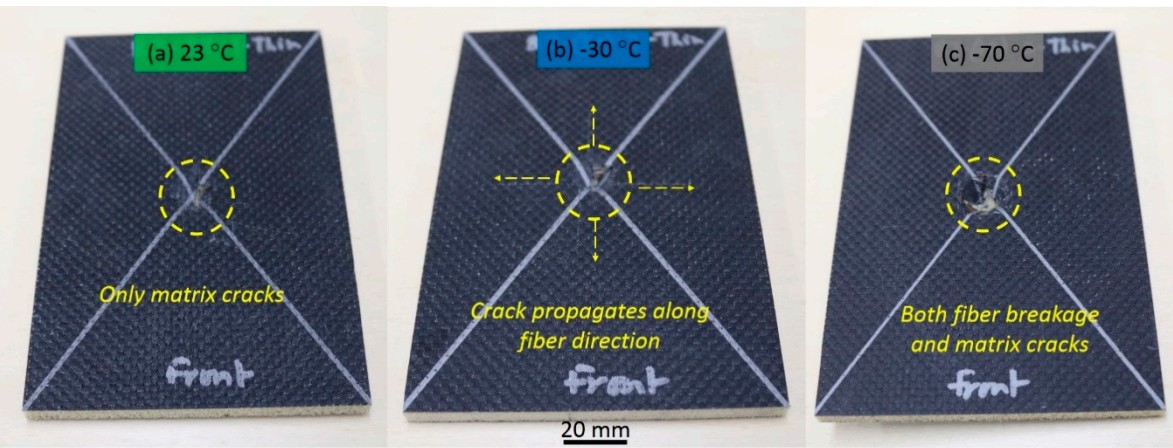

**Figure 6.** Impact damage (8 J) for thin face sheet sandwich composites at: (**a**) 23 °C; (**b**) −30 °C; (**c**) −70 °C.

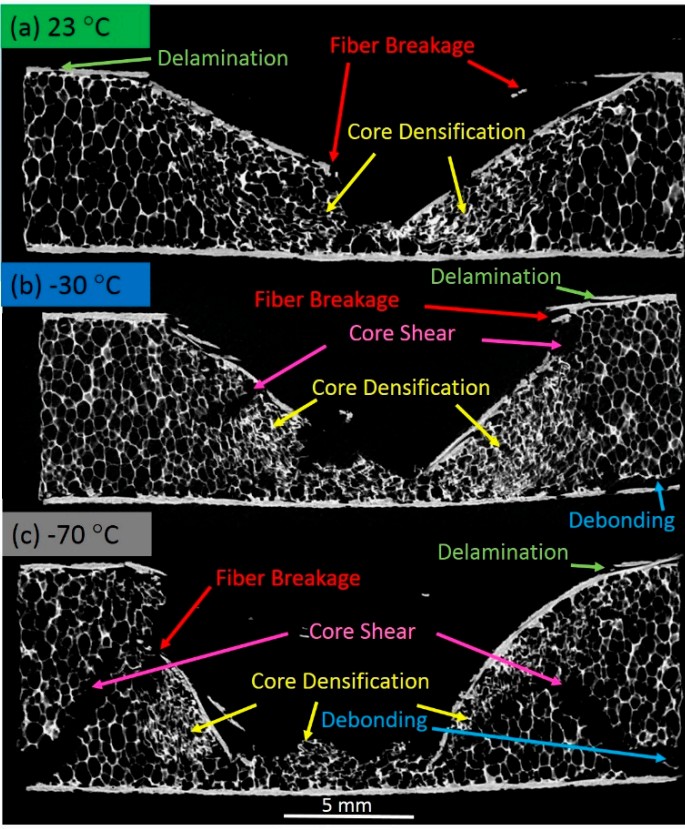

**Figure 7.** Post-impact *μCT* images of impact center for thin face sheet sandwich composites impacted with 8 J impact energy at (**a**) 23 °C; (**b**) −30 °C; (**c**) −70 °C.

As depicted in Figure 7a,b, core densification was dominated by the impacted front face sheet compression at 23 and −30 °C, whereas densification occurred near the bottom face sheet due to impactor tip compression (no severe back face sheet penetration) at −70 °C (Figure 7c). At −30 °C, core densification became prominent and led to the onset of core shearing (Figure 7b). Core shear is defined by the propagation of crack forming shear bands with a 45° angle across the core thickness. Thus, the shear bands propagate conically from the impact face sheet side into the foam core layer, as portrayed in Figure 7b,c. Core shear eventually results in debonding of the core from the back face sheet (Figure 7b,c). As the temperature decreases, foam core material becomes extremely brittle whereby cracks can propagate easily [22], and core shear stress increases [23], which subsequently leads to severe core shear failure at extreme low-temperature −70 °C (Figure 7c). The damage modes of core crushing and densification become more evident with the decrease of temperature. The debonding area also increases with the decrease of temperature in Arctic conditions as described by Elamin et al. [24]. It is worth mentioning that this study differs from [16] with the aim to relate core shear intensity to residual flexural strength after impact.

### 3.3. Residual Flexural Strength after Impact

The bending test results for post-impacted specimens compared to nonimpacted specimens are shown in Figure 8. The force-displacement curves were characterized by an initial linear elastic regime followed by a load drop. Nonimpacted specimens showed a trend of increasing peak force or flexural strength for decreasing temperature. The thick specimens broke catastrophically for bending outward cases at −70 °C (Figure 8c), while at other temperatures, the specimens typically exhibited a gradual decline in flexural force after the peak force. This is due to the embrittlement of the PVC core at low temperature −70 °C that facilitates core shear. Figure 9 plots the flexural peak load against pre-bending impact energy. Generally, when impact energy increases, flexural peak load decreases.

Bending outward cases perform better due to an undamaged top face sheet matrix, which can resist compressive load during bending. Although the bottom face sheet suffers impact damage, the damage is typically in the form of matrix cracking, thus the bottom face sheet can still withstand tensile load by the intact carbon fibers that are not severely damaged. However, for bending inward cases, the top face sheet has reduced compressive properties due to prior impact that damages the matrix. Therefore, the top face sheet was unable to withstand compressive stresses during three-point bending test and exhibited reduced flexural strength.

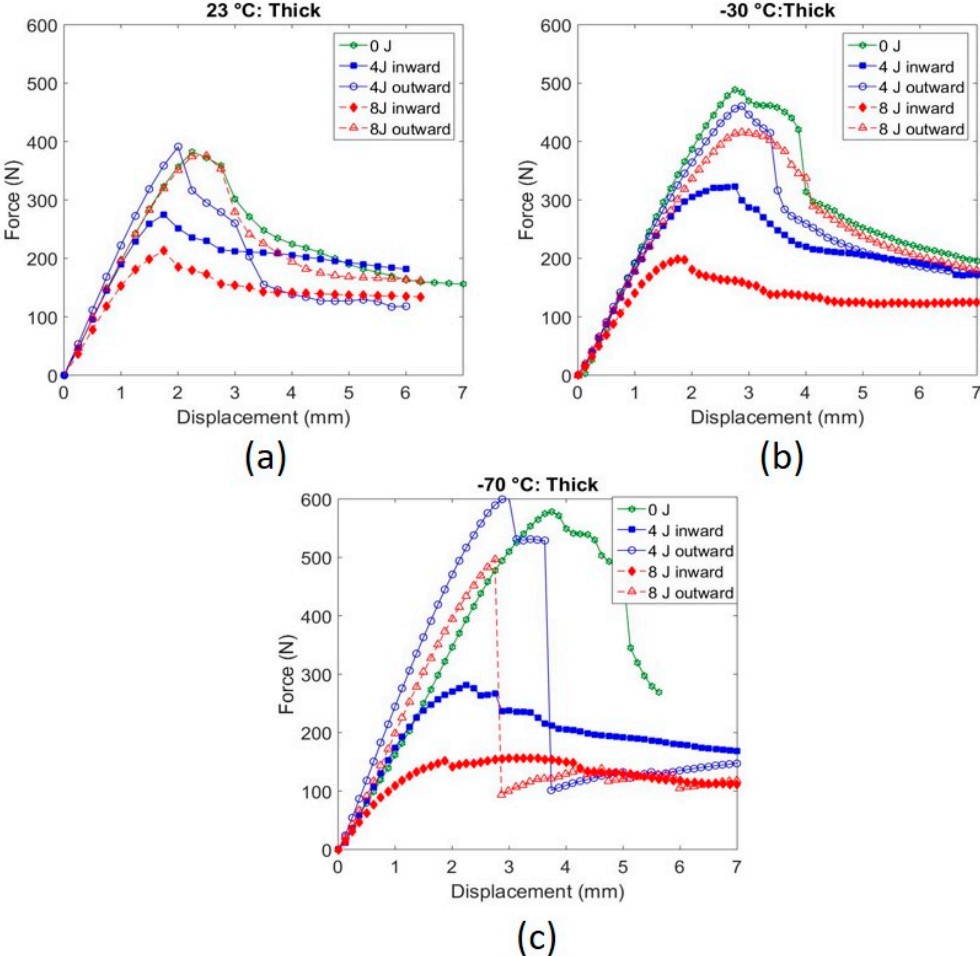

**Figure 8.** Force against displacement plots for thick specimens under three-point bending at (**a**) 23 °C; (**b**) −30 °C; (**c**) −70 °C.

Figure 10 shows the effect of temperature on flexural peak force of the nonimpacted beams. It is interesting to note that the bending behavior seemed to exhibit more "resistance" with decreasing temperatures. This is attributed to the fact that at low temperature, the compressive strength of the front CFRP face sheet increases, which makes the nonimpacted materials perform better under flexural load. Flexural strength is also found to be increased at Arctic subzero temperature for foam structures [25].

*3.4. Post-Bending Micro-Computed Tomography*

Figure 11 depicts the plan view of μCT images taken at 3.5 mm from the top face sheet, showing contrast in core shear intensity at impact and bending after impact cases. From the plan view, core shear appears as a ring of empty space with variable radius along the core thickness. This substantiates the previous discussion that at low temperature (−70 °C), prominent and severe core shearing occurs with an approximate diameter of 18.2 mm circulating the impact zone and gradually approaches to the back face sheet forming a 3-D conical shape (Figure 11a). However, after post-impact

bending the core shear front diameter has increased from 18.2 mm to 20.9 mm (Figure 11b), by a significant +15 %. This increase in core shearing is not associated with any new damage mechanisms. Rather, core shear front expansion occurs only due to flexural bending of the specimen.

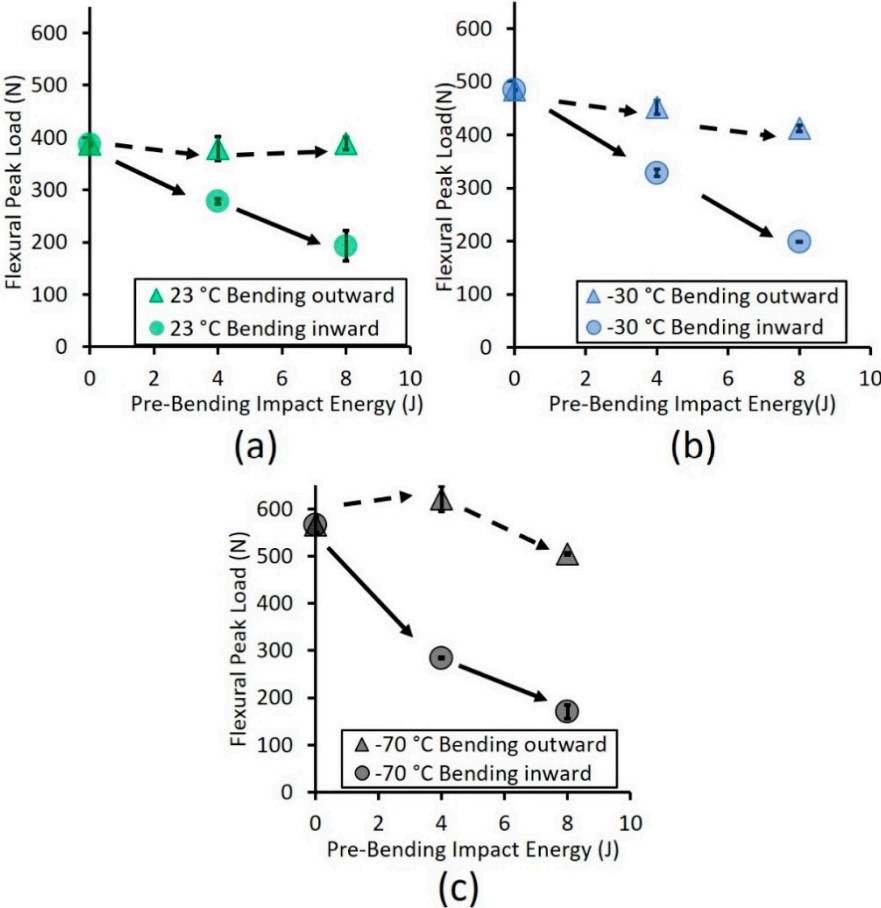

**Figure 9.** Effect of temperature on flexural peak load for thick impacted specimens at (**a**) 23 °C; (**b**) −30 °C; (**c**) −70 °C.

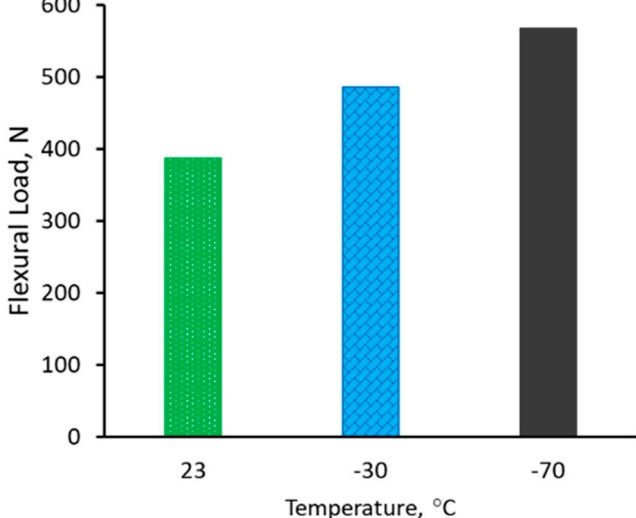

**Figure 10.** Flexural peak force against temperature for nonimpacted thick specimens at 23, −30 and −70 °C.

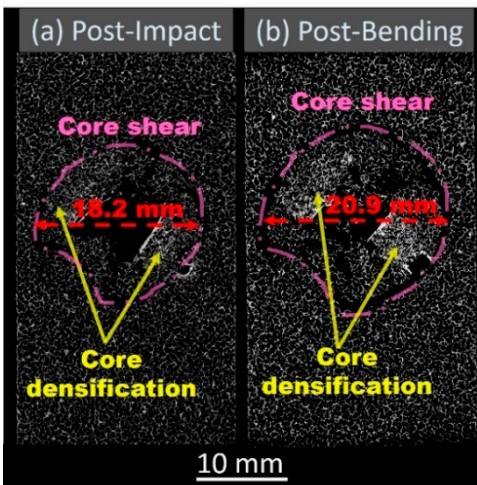

**Figure 11.** *μCT* images of foam core damage in thin face sheet composites (8 J, −70 °C): plan view at 3.5 mm from front face sheet for (**a**) post-impact; (**b**) post-bending condition.

*3.5. Statistical Analysis for Understanding the Influence of Factors*

The Pareto chart obtained from Minitab shows the absolute significance of the standardized effects from the biggest to the smallest. The reference line shows the effects which are statistically significant. Statistical software Minitab 18 yields the Pareto charts as shown in Figure 12a which indicates that face sheet thickness is the most significant factor that governs the flexural peak force, followed by pre-bend impact energy and temperature. Earlier observation in Figures 8 and 9 showed similar observations that these three factors were the prominent factors controlling flexural performance for the composite sandwich panel. Bending configuration had a much smaller effect than the other three.

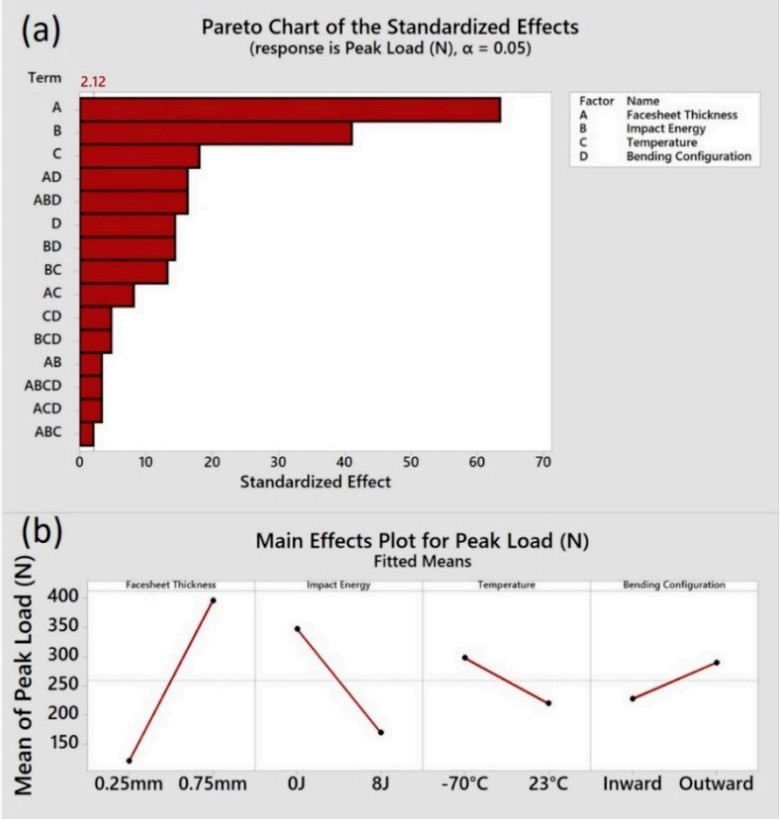

**Figure 12.** (**a**) Pareto chart; (**b**) main effects plot for post-impact flexural peak load.

The main effects plots, given in Figure 12b, which identified the influence of individual factors further validated the findings from the Pareto charts. The main effects plot provides the average data at the low and high factors. The gradient of the straight line corresponds to the significance of the factor directly. It is evident that the flexural peak force increases with increase in face sheet thickness (Th) whereas it decreases with increase in impact energy (Energ) and temperature. Inward bending shows inferior flexural performance compared to outward bending configurations. The flexural peak force is mainly influenced by face sheet thickness and impact energy whereas temperature and configuration (Conf) both have mild influences. Figure 13 portrays the interaction plot, which interprets the two-way interaction of the factors. If the slope of the two lines is not the same, there exists some interaction. It is concluded that apart from face sheet thickness and impact energy, which both are strong independent factors, all the other factors have interactions with each other.

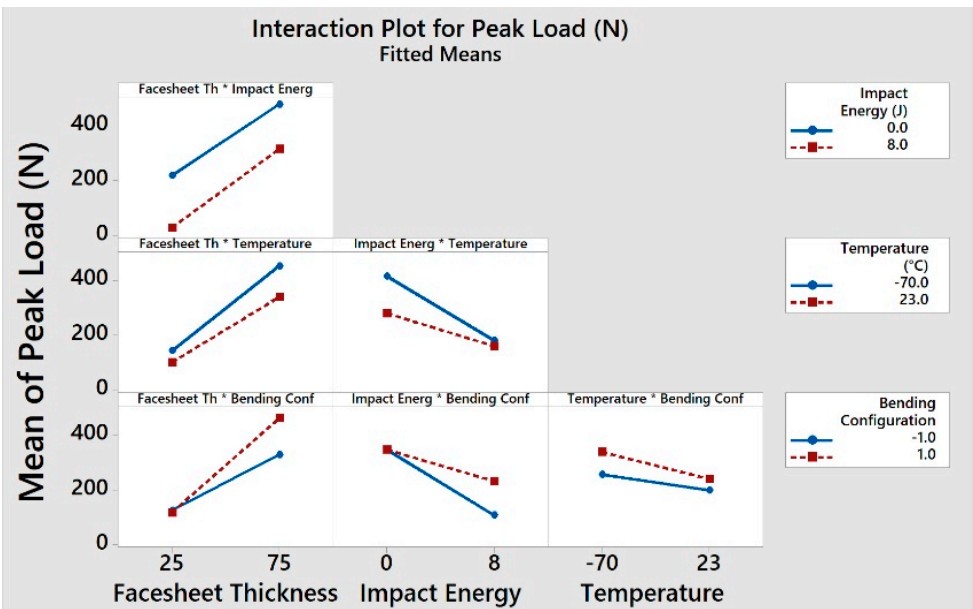

**Figure 13.** Interaction plot for flexural peak load, showing the interaction of face sheet thickness, impact energy and temperature on flexural force.

Contour plots are obtained by the constant response line projections from the response surface into the two-dimensional factor plane. Thus, the contour plot gives the prediction of impact energy and temperature on the x- and y-axes, respectively, along with contour lines for the peak load. At a specific impact energy, the contour plot can be used to graphically predict the flexural force for a particular temperature. Figure 14a shows that increasing impact energy will significantly reduce peak force for a given temperature for the bending inward case, whereas for the outward case, impact energy has a gentler influence (Figure 14b). This substantiates the experimental observation in Figure 9, as discussed earlier. Analysis of variance (ANOVA) is a statistical tool that calculates data based on the difference between two or more means. The variance within the groups and variances between the groups are compared. Subsequently, a regression equation can be obtained to describe the relationship between the response and the variables or factors. The algebraic representation of the regression equation for a linear model can be represented as:

$$Y = c + m_1 x_1 \tag{1}$$

where $Y$ is the response variable, $c$ is the constant or intercept of y-axis, $m_1$ is the slope of the line and $x_1$ is the value of the factor. The regression equation with more than one factor can be represented as:

$$Y = c + m_1 x_1 + m_2 x_2 + \ldots\ldots\ldots + m_n x_n \tag{2}$$

where $m_1, m_2, \ldots, m_n$ are the coefficients of the $x_1, x_2, \ldots, x_n$ factors. From ANOVA, considering the factors having *p*-values less than 0.05 only, the regression equation for the bending peak force, P is as follows:

$$
\begin{aligned}
P(N) = 75.91+ \quad & 4.743t - 23.29IE - 0.501 \times T - 0.000 \times C + 0.0951t \times IE \\
& -0.01906\,t \times T + 0.000 \times t \times C + 0.1060T \times IE - 0.000 \times T \times C \\
& -9.32\,C \times IE + 0.000965\,t \times T \times IE + 0.3160\,t \times C \times IE \\
& +0.0000T \times t \times C + 0.0225\,C \times T \times IE \cdot - 0.001553\,t \times T \times C \times IE
\end{aligned}
\tag{3}
$$

here, *t* is the face sheet thickness, *IE* is the impact energy, *T* is the temperature and *C* is the bending configuration. Graphical prediction based on contour plot and statistical prediction based on regression Equation (3) are further compared with experimental results for flexural peak force for 4 J pre-bend impact energy at 23, −30 and −70 °C, as shown in Figure 15. Excellent agreement between experimental data and statistical prediction was observed. The greatest differences were for the −70 °C cases, where the statistical predictions deviate from experimental data. In reality, the impacted front face sheet matrix which was in compression during inward bending was severely embrittled at −70 °C [22,23] that led to flexural failure of the specimen at a much lower force. However, for the outward case, linear assumptions made by the statistical analysis underpredicted the experimental peak load, whereby the undamaged front face sheet compressive properties were enhanced at low temperature (−70 °C) and bottom face sheet tensile properties were uncompromised due to little or no damage during impact.

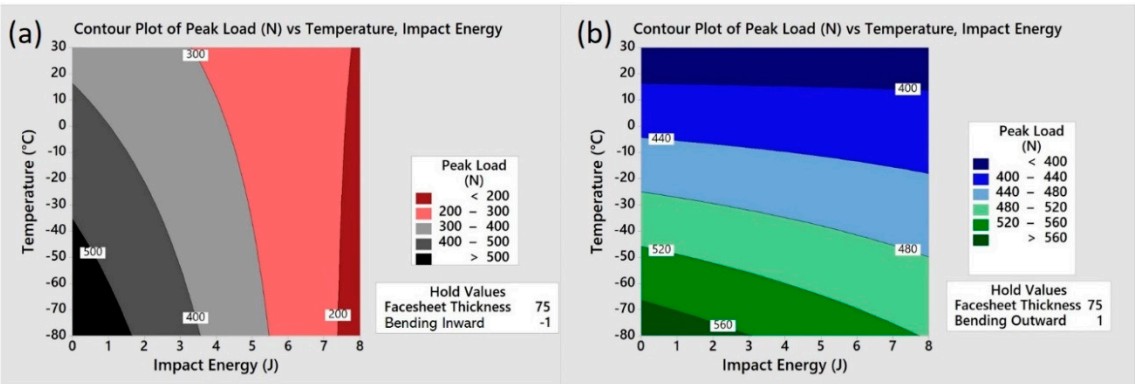

**Figure 14.** Contour plot prediction of flexural peak load for 0.75 mm face sheet sandwich composites under (**a**) bending inward; (**b**) bending outward configuration.

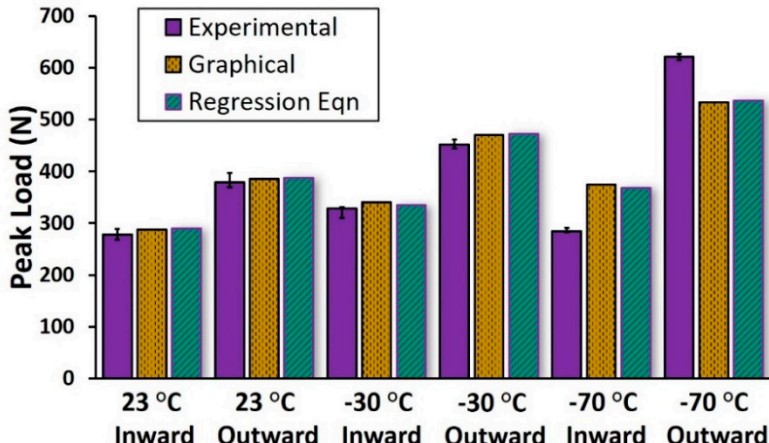

**Figure 15.** Statistical analysis and prediction of post-impact flexural peak load for 0.75 mm face sheet sandwich composites under 4 J impact energy.

## 4. Conclusions

The impact performance and bending behavior of carbon-fiber foam-core composite sandwich structures in low-temperature Arctic conditions were investigated. Impact performance is affected by impact test temperature and drastically lowered at extremely low-temperature Arctic conditions. At 23 °C, matrix cracking and delamination are dominant damage mechanisms; while at −70 °C, core shear, debonding, fiber breakage and severe face sheet penetration are the dominating failure modes. The intensity of core shear and debonding increases with decrease in temperature. Post-impact flexural strength generally decreases with increase in pre-bend impact energy. Flexural performance is superior in the case of bending outward rather than in the case of bending inward, due to reduced compressive properties at the impacted face. Flexural damage mechanisms associated with post-impact and post-bending conditions at −30 and −70 °C are dominated by core shear and debonding. Statistical techniques were employed to investigate flexural peak force by understanding the effect of the factors. It was revealed that flexural performance was mainly driven by temperature, face sheet thickness and pre-bend impact induced damage or impact energy. Graphical contour plots and regression equation were derived for predicting the bending collapse or peak force and further compared with experimental data. A good agreement was achieved between experimental and statistical results. The findings from this work provided a better understanding on the impact behavior and post-impact flexural performance of carbon-fiber foam-core sandwich composites when employed in extremely cold Arctic temperatures.

**Author Contributions:** Conceptualization, M.H.K. and K.T.T.; methodology, M.H.K.; validation, B.L. and K.T.T.; investigation, M.H.K.; resources, K.T.T.; data curation, M.H.K., B.L. and K.T.T.; writing—original draft preparation, M.H.K.; writing—review and editing, B.L. and K.T.T.; visualization, M.H.K.; supervision, K.T.T.; project administration, K.T.T.; funding acquisition, K.T.T. All authors have read and agreed to the published version of the manuscript.

**Funding:** This research was funded by the U.S. Office of Naval Research (ONR Solid Mechanics Program Manager: Yapa Rajapakse), grant number N00014-18-1-2546.

**Acknowledgments:** The authors would like to thank Lingyan Li for her help with the X-ray micro-CT scanning of the specimens.

**Conflicts of Interest:** The authors declare no conflict of interest.

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
