# Peer review of "Impact Performance and Bending Behavior of Carbon-Fiber Foam-Core Sandwich Composite Structures in Cold Arctic Temperature"

_jcs, doi:10.3390/jcs4030133_

Round 1

Reviewer 1 Report

This is a nice contribution. Perhaps a couple of references on similar recently published work could be added for completeness: https://doi.org/10.1016/j.compositesb.2018.06.035

Reviewer 2 Report

  • General: the phrase “Cold Temperature Arctic Conditions” indicates specific conditions in addition to the low temperature. If there are: please specify, if not: “arctic conditions” is an obsolete phrase an should be removed
  • 2.1. Test Specimens:
    • Please give more detailed information on the investigated material. Please specify type of weave, type of epoxy resin, type of fibre, manufacturing process for the “woven carbon fiber epoxy matrix face sheets” and the “composite sandwich panels” with the related process parameters, bonding procedure and interface condition.
    • The statement “For low-temperature testing, specimens are conditioned in a freezer at - 23 °C for a day to achieve thermal equilibrium.” Is very unclear. Please explain more detailed why this has been done or correct. Does this means that if tests are carried out at -30 and -70 °C no thermal equilibrium is achieved.
  • 2.2. Impact Test: How are the forces and deflections are measured?
  • 2.5. Statistical Design of Experiment:
    • Please provide the final testing matrix containing number of test repetitions containing how many specimen were chosen for which configuration and for “inward”, “outward” and “non-impacted” post impact bending.
    • Does “Impact energy 0 J” in Table 1 refers to the “non-impacted” bending configuration? Allthough it becomes a bit clearer later in the manuscript, it is not explained well here and should be improved. (“… two impact energy (4 J and 8 J) ... configurations … are considered…”)
  • 3.1. Impact Performance: The statement “Moreover, with decrease in temperature, the stiffness and displacement of force-displacement generally increase.” is not convincingly supported by Figure 2. Please correct.
  • 3.3. Residual Flexural Strength After Impact: The results are interesting, especially the fact that the bending behaviour seems to exhibit more “resistance” with decreasing temperatures. Please elaborate a bit more on this with an additional graph (e.g. peak force vs. temperature) and some statements.
  • 4. Conclusions: To the understanding of the reviewer, the statement “It is revealed that flexural performance is mainly driven by face sheet thickness and pre-bend impact energy.” is in conflict with Figure 7, which indicates a strong dependency of the peak force from the temperature. Also, it is questionable, whether this observation has any scientific value, since it is more or less trivial that thicker material withstands more load and more damage due to higher impact energies lead the less bending resistance. Especially given that the paper focuses on the temperature dependency, the above-mentioned statement might therefore be misleading. Please consider more comprehensive formulations and a more critical view on the meaningfulness of the performed statistical analysis (also in 3.5. Statistical Analysis for Understanding the Influence of Factors).

Reviewer 3 Report

The authors have tried to touch upon an important topic pertaining to impact behavior of sandwich composites in arctic conditions, however, their paper lacks a few technical details.

All figures, references and tables need to be cross-referenced in the text.

All figure captions must be descriptive

All figures must have good resolution

All the font sizes of the texts within figures must be readable and consistent and should not be too large.

In addition, Please address the comments in the attached draft.

Round 2

Reviewer 3 Report

The authors have made the changes suggested by the reviewers and the manuscript now looks much more improved.